# Effect of Deployment and Harvest Date on Growth and High-Value Compounds of Farmed *Alaria esculenta*

**DOI:** 10.3390/md21050305

**Published:** 2023-05-17

**Authors:** Silvia Blanco, Maria Sapatinha, Mick Mackey, Julie Maguire, Simona Paolacci, Susana Gonçalves, Helena Maria Lourenço, Rogério Mendes, Narcisa Maria Bandarra, Carla Pires

**Affiliations:** 1Bantry Marine Research Station, Gearhies, Bantry, P75 AX07 Co. Cork, Ireland; 2IPMA, IP, Instituto Português do Mar e da Atmosfera, DMRM, DivAV, Av. Dr. Alfredo Magalhães Ramalho, 6, 1495-165 Lisboa, Portugal; m.sapatinha@gmail.com (M.S.); sgoncalves@ipma.pt (S.G.); helena@ipma.pt (H.M.L.); rogerio@ipma.pt (R.M.); narcisa@ipma.pt (N.M.B.); cpires@ipma.pt (C.P.); 3Indigo Rock Marine Research Centre, Gearhies, Bantry, P75 AX07 Co. Cork, Ireland; m.mackey@indigorock.ie (M.M.); jmaguire@bmrs.ie (J.M.); sip@aquabt.com (S.P.); 4Centro Interdisciplinar de Investigação Marinha e Ambiental (CIIMAR), Terminal de Cruzeiros de Leixões, Av. General Norton de Matos s/n, 4450-208 Matosinhos, Portugal

**Keywords:** brown seaweed, bioactive compounds, phenolic compounds, nutraceuticals

## Abstract

*Alaria esculenta* is a brown seaweed farmed in many European countries for its biomass rich in useful bio compounds. This study aimed to identify the optimal growing season to maximise biomass production and quality. The seeded longlines of the brown seaweed were deployed in the southwest of Ireland in October and November 2019 and samples of the biomass were harvested in different dates, between March and June 2020. Biomass gain and composition, phenolic and flavonoid content (TPC and TFC) and biological activities (antioxidant and anti-hypertensive activities) of seaweed extracts prepared with Alcalase were evaluated. The biomass production was significantly higher for the line deployed in October (>20 kg·m^−1^). In May and June, an increasing amount of epiphytes was observed on the surface of *A. esculenta*. The protein content of *A. esculenta* varied between 11.2 and 11.76% and fat content was relatively low (1.8–2.3%). Regarding the fatty acids profile, *A. esculenta* was rich in polyunsaturated fatty acids (PUFA), especially in eicosapentaenoic acid (EPA). The samples analysed were very rich in Na, K, Mg, Fe, Mn, Cr and Ni. The content of Cd, Pb Hg was relatively low and below the maximum levels allowed. The highest TPC and TFC were obtained in extracts prepared with *A. esculenta* collected in March and levels of these compounds decreased with time. In general, the highest radical scavenging activities (ABTS and DPPH), as well as chelating activities (Fe^2+^ and Cu^2+^) were observed in early spring. Extracts from *A. esculenta* collected in March and April presented higher ACE inhibitory activity. The extracts from seaweeds harvested in March exhibited higher biological activity. It was concluded that an earlier deployment allows for maximising growth and harvest of biomass earlier when its quality is at the highest levels. The study also confirms the high content of useful bio compounds that can be extracted from *A. esculenta* and used in the nutraceutical and pharmaceutical industry.

## 1. Introduction

Seaweed cultivation has been a longstanding practice in Asia for decades, as documented by Cheng (1969) [1]. In contrast, the commercial cultivation of seaweed in Europe has occurred only in recent years [2,3]. Between 1950 and 2019, there was a significant increase in brown seaweed farming worldwide, with production growing from 1.3 to 16.4 million tonnes. In 2019, brown seaweed represented 47.3% of the world’s seaweed cultivation [4]. Over time, the industrial utilisation of seaweed biomass has evolved from using beach-cast seaweeds as fertilisers and linen bleaching [5,6], to employing them as sources of natural bioactive compounds due to their strong antioxidant potential [7].

Seaweed is often subjected to challenging environmental conditions, which results in its ability to produce a wide range of metabolites such as enzymes, pigments, polysaccharides, vitamins, phenolics, tocopherols, etc., that protect them from external stressors [8,9,10,11,12]. Brown seaweed possesses high levels of phenolic compounds including phenolic acid, flavonoids, and tannins. These compounds are mainly responsible for the high activity of brown algae [13] and play vital roles in cell-wall synthesis and storage, herbivore deterrence, stress response and protection against UV-radiation [14]. However, other seaweed compounds such as carotenoids (fucoxanthin and astaxanthin), sterols, carbohydrates, and vitamins (E and C), proteins (phycobiliproteins) and peptides or amino acids (mycosporine-like amino acids) also show antioxidant properties [15]. Brown algae also contain bioactive compounds such as phlorotannins and fucoxanthin, with anti-diabetic capacity through the inhibitory effect of enzyme targets such as α-amylase and α-glucosidase [16]. Moreover, it was shown that peptides obtained by enzymatic hydrolysis of seaweed proteins exhibited angiotensin-converting-enzyme (ACE) inhibitory activity and can be used as hypotensive commercial products [17]. Additionally, phlorotannins bound to proteins and released during enzymatic hydrolysis are also excellent ACE inhibitory compounds [18].

*Alaria esculenta*, or Atlantic wakame, is a common edible brown seaweed found on exposed rocky shores of the Atlantic Ocean in northern Europe and America [19]. *A. esculenta* (along with *Saccharina latissima*) is one of the most cultivated marine algae in Europe. As it is fast-growing species, accounting for a yearly production of 24 and 44 tons of fresh weight in Ireland and Norway respectively—the largest producers of *A. esculenta* in Europe [20]. Nonetheless, the harvesting time is well documented as determining the biochemical composition of seaweeds, influencing the content of bioactive compounds that could be extracted [21]. Polyphenolic compounds such as phlorotannins and polysaccharides like fucoidan are known to show seasonal variations, which become important to determine the most favourable time for the harvest of seaweed as raw material for nutritional supplements or for pharmaceutical purposes [22].

While seasonal variation in biomass quality is documented for some species, there are no studies focusing on the identification of the most appropriate growing season for *A. esculenta*.

The object of the present study was to identify the best growing window (in terms of productivity and biomass quality) for the farming of a strain of *A. esculenta* in the southwest of Ireland. Thus, the aim of the study is to improve the management of *A. esculenta* production, by assessing the influence of the date of deployment of seeded long-lines on the biological activities (antioxidant, anti-diabetic, anti-hypertensive, and antimicrobial) of the algae.

## 2. Results and Discussion

### 2.1. Production

Samples of *A. esculenta* fronds from the lines deployed on different days were taken at different points in time, between March and June 2020, to assess the increase in biomass (Figure 1). The average monthly sea temperature was 13.8 and 11.5 °C in October and November, respectively. The sea temperature decreased constantly until March 2020. After this month, a gradual increase was registered, reaching 13.6 °C in June 2020.

On 20 March (first sampling date), line 2 (deployed around 6 weeks before line 7) had an average biomass of 11.92 ± 1.09 kg·m^−1^, while the biomass carried by line 7 was 7.9 ± 0.95 kg·m^−1^. During the three following sampling dates, there was no significant increase in biomass in any of the two lines. On the last sampling day (June 6th) an increase in biomass was observed (20.66 ± 1.31 kg·m^−1^ for line 2 and 14.37 ± 4 kg·m^−1^ for line 7); however, this increase was statistically significant only for line 2 (*p* = 0.03). The line deployed earlier produced a significantly higher amount of biomass than the line deployed on 28 November (*p* = 0.02). In May, a small number of epiphytes started colonizing the surface of *A. esculenta* in both lines observed, the following month the number of epiphytes had increased to a point to compromise the quality of the kelp biomass.

Kerrison et al. [23] observed the biomass gain and the length of *A. esculenta* in the Port a’ Bhuiltin seaweed farm, in Scotland, every month from deployment (20 February 2016) to harvest (15 August 2016). The final biomass they reported, 4.5 kg·m^−1^, is considerably lower than the biomass observed in the present study. The authors reported a decreased growth after 19 June, with fronds biomass non-changing significantly in the last two months. The late deployment can explain the reduced biomass gain.

A very diversified biomass yield for *A. esculenta* was reported by Kraan [19]. The author observed the growth of five different North Atlantic strains in Ard Bay (Ireland) and reported that a Canadian strain produced over 45 kg m^−1^ of biomass. A hybrid strain produced by crossing an Icelandic and an Irish strain produced 13.75 kg wet weight m^−1^, while an Irish strain from the Aran Islands, Ireland, produced 7.4 kg m^−1^. Reis et al. [24] obtained a yield of 10 kg m^−1^ in south Galway Bay, Ireland. The heterogeneity of biomass yields highlights the importance of strain selection for kelp farming.

In March 2020, the average length of fronds (Figure 2) was 285.3 ± 56.6 cm for line 2 and 224.7 ± 8.7 cm for line 7. The difference between the two lines was not statistically significant. At the beginning of April, the average length of fronds in line 2 was significantly higher (*p* < 0.01) than in line 7 (356.7 ± 13.9 cm vs. 224.4 ± 21.4 cm). The average length of fronds in line 7 stayed constant until May, while fronds in line 2 were subjected to erosion and at the end of April their average was 281.8 ± 11.5 cm. In June, a further decrease in length was observed for both lines, with the average length being 243.0 ± 11.2 cm in line 2 and 167.4 ± 30.1 cm in line 7. Erosion of kelp is the loss of tissue from the fragmentation of distal parts of the fronds [25] and it is a well-known phenomenon described by several authors [26,27,28,29]. The length of fronds observed in the present study is also considerably higher than the data reported by Kerrison et al. [23]. These authors observed a maximum length of fronds of 160 cm, at the end of May, in their seaweed farm in Scotland, with lines deployed on 20 February 2020. The length of the fronds decreased in June and continued to decrease until August because of erosion.

### 2.2. Characterisation of Dried A. esculenta–Proximate and Mineral Composition and Heavy Metals

Samples deployed on 15 October (line 2) and on 28 November (line 7) of 2019 were analysed as a single pooled sample of those harvested on 20 March, 3 April, 22 and 30, 5 May and 2 June of 2020. In total, eleven parallel lines were deployed that year. The first and the last line were excluded as they both had an unprotected side that could potentially affect the growth of the seaweed making the results less representative of the majority of the lines. Line 2 and line 7 were chosen because they were reasonably distant from each other to guarantee independence. The proximate composition of these samples is shown in Table 1. Protein content varied between 11.2. and 11.8%. The highest protein content was recorded in samples deployed earlier. The protein contents found in this study seem to be generally in agreement with those reported in other studies, and determined with different methods, for the same species. For example, Schiener et al. [30] measured a protein content, in *A. esculenta*, of 11.0 ± 1.4% using Lowry assay. Mæhreet al. [31] analysed the content of amino acids using an amino acid analyser that quantified these molecules chromatographically. The authors calculated the protein content from the sum of individual amino acid residues and found a content of 10.7% in *A. esculenta*. Stévant et al. [32], using the same methodology used in the present study (determination of nitrogen content and use of conversion factor of 5), determined a protein content of 10.5%.

A higher protein content in brown seaweed sampled in winter months was observed by other authors [33,34]. Chapman and Craigie [35] suggested that kelp tends to build up nitrogen reserves in winter months to sustain growth in spring. In the present study, the protein content refers to homogenised samples collected between March and June, hence the effect of seasonality on protein content cannot be appreciated. However, it is reasonable to speculate that the line deployed later was exposed for less time to the autumn conditions that promote nitrogen accumulation, and this is reflected in the total protein content of the homogenised samples. The lipid content in lines deployed on different dates was also significantly different. Biomass from line 2 (deployed in October) had a lipid content of 2.3 ± 0.2 g per 100 g of biomass, while biomass from line 7 (deployed in November) contained 1.8 ± 0.2 g per 100 g of biomass. These lipid levels are in accordance with other studies on *A. esculenta*.

Regarding the fatty acids profile, as shown in Table 2, *A. esculenta* samples are rich in PUFA followed by saturated fatty acids (SFA) and monounsaturated fatty acids (MUFA). Both samples were rich in EPA, but the levels of docosahexaenoic acid (DHA) were relatively low (<1%). Palmitic acid (16:0) is dominant but the *A. esculenta* samples have also significant levels of oleic (18:1 ω9) and arachidonic acid (AA, 20:4 ω6).

AA influences cell membrane fluidity and the activity of ion channels, especially in the brain, it represents 20% of the human brain’s dry weight, together with DHA. Moreover, it is also recommended for the balance of the central nervous system [36].

Low SFA contents and higher contents of MUFA and PUFA observed in the present study confirm the benefits of consuming this seaweed for human health as replacing SFA in the diet with MUFA or PUFA is beneficial to the cardiovascular system [37]. EPA and DHA are of interest to the nutraceutical industry because of their role in visual and cognitive development and function. EPA and DHA are also beneficial to inflammatory-related diseases [38]. EPA content in *A. esculenta* in this study agrees with previous observations [31,39,40].

Omega-3 fatty acids have anti-inflammatory activity, and they are documented to reduce the incidence of cardiovascular diseases [38,41]. A high ω3/ω6 ratio in the biomass is considered a positive characteristic as modern diet in industrialised countries is generally too rich in ω6 [42]. The ω3/ω6 ratio in this study is in accordance with the ratio observed for *A. esculenta* by Foseid et al. [43].

Variability of the fatty acid profiles in seaweed depends both on genetic predispositions and environmental factors. Seasonality can affect the fatty acid profile as it is linked to changes in irradiance, salinity, nutrients availability, and the presence of epiphytes [44,45,46,47,48]. However, in this study, no significant difference was observed in the fatty acids profile of fronds deployed in October and fronds deployed in November.

The ash content gives a rough estimate of the total mineral content in the biomass. The ash content was relatively high in both *A. esculenta* samples reflecting the total mineral content, reaching approximately 32% of the dry weight of the biomass and *A. esculenta* from line 2 had significantly higher ash content than that of line 7 (Table 1). These values are close to levels reported in previous studies for the same species [30,31,32]. A more detailed mineral composition of *A. esculenta* is presented in Table 3. The data are discussed here mainly from a human consumption perspective because the levels of these elements suggest both risks and potential benefits from consuming seaweeds. As can be seen, the high ash content is due to the accumulation of mainly potassium and sodium ions. Thus, both *A. esculenta* samples (Line 2 and 7) were rich in Na and K, however, they showed Na/K ratios below 1.0. The *A. esculenta* samples were very rich in Mg, Fe, Mn, Cr and Ni and the sample deployed on line 2 presented significantly higher values of these minerals except Mg.

In general, the Mg, Cu, Mn and Zn content of *A. esculenta* samples showed the same magnitude with the exception of Fe as that of *A. esculenta* observed by Mæhre et al. [31]. The data obtained in this study are also similar to that reported by Olsson et al. [50] in 9 species of brown seaweeds from the Swedish west coast. Considering the DRI, these *A. esculenta* samples are a good source of several minerals. For example, to attend the DRI of iron of 8 mg/day one would have to ingest about 20–56 g of dried seaweed.

Algae are known for their ability to accumulate heavy metals with adverse health effects. Thus, in this study, the amounts of Cd, Pb and Hg were analysed (Table 4). No significant differences were observed between the content of these metals in the samples deployed on lines 2 and 7.

Considering dried seaweeds as a food supplement, the maximum levels allowed by the EU [51,52] for cadmium and lead are 3 mg/kg. These limits were not exceeded in both *A. esculenta* samples. In what concerns Hg, the levels observed in *A. esculenta* samples were very low (<0.011 mg/kg) and under the recommended limit for this contaminant (0.10 mg/kg) by the EU [53]. According to the regulations, *A. esculenta* samples were suitable for consumption.

Cd levels of *A. esculenta* were significantly lower than that reported by Mæhre et al. [31] for Norwegian *A. esculenta*. However, the results presented in this study were in the range of those observed for other Norwegian seaweed species (0.091–3.8 mg/kg dw) [31] and brown seaweeds of the Swedish west coast (0.06–1.46 mg/kg dw) [50]. *A. esculenta* samples also showed similar Cd levels (considering the water content of around 80%) to that of seaweeds collected from different regions in Korea [54].

The levels of Pb obtained in this study were in the range of those observed for Swedish brown seaweeds which varied between 0.05 and 10 mg/kg dw [50] and those reported by Lee et al. (0.025–0.222 mg/kg ww) [54].

Concerning Hg, the levels observed in *A. esculenta* samples were below the limit of quantification which agrees with the results obtained for Norwegian *A. esculenta* [31] and Swedish brown seaweeds [50].

It is known that the levels of these contaminants in seaweeds vary with factors such as seasonal variations, algal growth conditions (size, age, and nutritional state), environmental factors (temperature, salinity, light and pH) and the presence of industries or cities nearby [55].

### 2.3. Extraction Yield, Total Phenolic and Flavonoids Content (TPC and TFC)

Extraction yields, calculated as the weight percentage of lyophilised extract to the weight of dried seaweed submitted to extraction are shown in Figure 3A. The Alcalase-assisted extraction yields varied between 31.8 ± 2.5 and 44.0 ± 5.0% and in general, no significant differences were obtained between extracts prepared from *A. esculenta* deployed on different dates. Extraction yields obtained in this study were similar to those reported by several authors for Alcalase-assisted extraction [56,57]. However, previous studies with *A. esculenta* harvested in Bantry Bay (Cork, IE) showed lower yield values when Alcalase was used (19.7%) [58].

Phenolic compounds are antioxidant secondary metabolites found in seaweeds and these compounds vary from simple molecules such as phenolic acids to highly complex compounds. This class of compounds also includes flavonoids, a big group of secondary metabolites. In this study, the TPC of *A. esculenta* ranged between 57.4 ± 1.9 and 120.0 ± 1.8 mg GAE/g dried extract and decreased from March to June (Figure 3B). The highest TPC was obtained in the extract prepared with *A. esculenta* deployed on 28 November (Line 7) and collected on 20 March. The extract prepared with *A. esculenta* deployed on line 7 and harvested on 12 May showed the lowest TPC. There were no significant differences between the TPC of *A. esculenta* of the two different lines.

Flavonoids are another subgroup of phenolic compounds functioning as UV screen in plants. The TFC extracts of *A. esculenta* showed values between 15.5 and 29.1 mg QE/g dried extract and the highest values were observed with extracts prepared with *A. esculenta* harvested on 20 March (Figure 3C).

It has been reported that the antioxidant content in seaweeds shows seasonal variation and that the concentration of pigments is generally lower in summer and higher in winter [59]. This depends on changes in light availability and nutrient concentrations in the water [60]. Increased pigments are usually associated with low light availability [61] as the pigments increase light-harvesting efficiency. Moreover, biotic factors such as grazing pressure, reproductive state, and stage within the life cycle have been correlated with changes in the polyphenols content [62,63]. The present study confirms a significant decrease in TPC and TFC in March and this is probably linked to the increase in light availability in this month, at the latitude where the seaweed farm is located. In February 2020, the southwest coast of Ireland received solar radiation of 12,272 Joules·cm^2^ [64]. In March, the radiation received was more than double (27,703 Joules·cm^2^). This increased light is probably the reason for the decrease in TPC and TFC observed. A further increase in solar radiation in April (46,503 Joules·cm^2^) was reflected on the TPC, but not on the TFC.

The radical scavenging activity of samples deployed and collected on different days was assessed. Regarding ABTS radical scavenging, extracts of *A. esculenta* showed EC_50_ values between 3 and 6 mg·mL^−1^, and the lowest values were observed with extracts prepared with *A. esculenta* collected on 20 March and 3 April (Figure 4A). ABTS radical scavenging capacity of seaweed is related to their phenolic hydroxyl group [65]. DPPH radical scavenging activity was also assessed. DPPH is a compound that possesses a nitrogen-free radical. A free radical scavenger destroys DPPH when interacting with it. This assay was used to test the ability of the antioxidative compounds to function as proton scavengers or hydrogen donors. The EC_50_ was determined to quantify the radical scavenging effects. The lowest value of EC_50_ indicates the strongest ability of the extract as DPPH scavengers. In this study, some extracts did not attain 50% of inhibition for the concentration range tested, and the highest activity was obtained with *A. esculenta* collected on 3 April (Figure 4B).

The ferric reduction assay was also carried out to detect the presence of antioxidants in the sample. These cause the conversion of the Fe^3+^/ferricyanide complex into the ferrous form in a redox-linked colorimetric reaction. The reducing power of *A. esculenta* extracts is shown in Figure 4C. The highest absorbances were measured in the extract prepared from *A. esculenta* collected on 20 March and deployed on line 7.

Overall, the highest antioxidant activity was observed in the extracts prepared with *A. esculenta* collected on the first days of sampling, when the highest phenolic and flavonoid content were also observed. A correlation between radical scavenging activity with phenolic and flavonoid content was observed by other authors. For example, Farasat et al. [66] observed that the total phenolic and flavonoid content in Ulva clathrate showed a positive correlation with the DPPH radical scavenging activity and negative correlations with IC_50_.

The Cu^2+^ chelating capacity of the different extracts prepared is shown in Figure 5A. The lowest EC_50_ values were recorded in extracts from *A. esculenta* collected on 20 March (2.36 and 2.40 mg/mL for lines 2 and 7, respectively). On the other hand, the extract prepared with *A. esculenta* collected on 12 May (line 7) presented the lowest Cu^2+^ chelating activity.

Regarding Fe^2+^ chelating activity (Figure 5B) no noticeable trend was observed with the sampling dates, but it is possible to see that the extracts from the deployment line 2 had in general lower EC_50_ values. As Cu^2+^ chelating capacity, Fe^2+^ chelating activity was also higher in the case of the extract prepared with *A. esculenta* collected on 20 March.

Concerning ACE inhibitory activity, the extracts with higher ACE inhibitory activity (*ca* 50%) were those of 20 March and 22 April. On the contrary, extracts prepared with *A. esculenta* collected on 12 May and 2 June showed the lowest ACE inhibitory activity. Moreover, in these extracts, no significant differences were observed between the two deployed lines (Figure 6).

As a conclusion, extracts prepared with *A. esculenta* deployed on 15 October and harvested on 20 March have higher biological activities.

### 2.4. Principal Components Analysis

A multivariate analysis was carried out on all data obtained from antioxidant and ACE activities of algae extracts to detect relationships between groups of samples (Figure 7). The first principal component (PC1, 49.04% of the total explained variance) is strongly correlated with TPC (loading 0.97), ABTS (−0.87), TFC (0.85), ACE (0.78) and Cu^2+^ (−0.72). The second PC (PC2, 18.14% of the total explained variance) is correlated with Fe^2+^, RP and DPPH (loadings were—0.75, 0.61, −0.50, respectively).

The plot (PC1 vs. PC2) shows a clear separation of extracts prepared from the different lines based on PC2, especially on Fe^2+^ chelating activity. Extracts prepared from *A. esculenta* deployed on line 2 had higher Fe^2+^ chelating activity than those prepared from line 7.

This plot also shows a clear separation of extracts prepared from the different sampling dates based on PC1, especially on TPC, TFC, ABTS radical scavenging activity and ACE inhibitory activity. Extracts prepared from *A. esculenta* sampled on 20 March and April (3rd and 20th) (lines 2 and 7) had higher TPC, TFC and ABTS radical scavenging activity. The findings suggested by PCA were supported by ANOVA.

The proximity of the projection of variables in the plot of principal components (PC1 and PC2) suggests a correlation between TPC and TFC (r = 0.75), TPC and ABTS (r = −0.86), TPC and Cu^2+^ chelating activity (r = −0.72) and TPC and ACE (r = 0.70). TFC is correlated with ABTS (r = −0.70) and with ACE inhibitory activity (r = 0.81).

## 3. Materials and Methods

### 3.1. Seed Preparation

*A. esculenta* was cultivated based on the manual ‘Cultivation of the brown seaweed *Alaria esculenta’* [67]. Mature sporophylls were collected in spring 2018 and 2019 from a natural bed in Gearhies, Bantry (Ireland; 51°38′49.8″ N 9°34′49.5″ W). A total of 40 sporophylls were collected each year from 20 mature individuals, cleaned and placed in an empty 2 L beaker, and kept in the dark at 10 °C. After 18–20 h, 1.5 L of distilled water was added to the beaker and stirred with a sterile stirring rod. The sporulation started immediately and continued for one hour. The solution containing the released spores was filtered through a 250 μm filter to remove debris and, subsequently, through a 60 μm filter. A 350 mL aliquot of the solution was then inoculated in a 4 L Pyrex flask containing a modified version of the culture medium formulated by Provasoli et al. [68] and prepared with autoclaved, UV and sand-filtered seawater. The flask inoculated with spores was placed initially at 10 °C and the temperature increased by 1 °C per day until it reached 14 °C. The spores were maintained at this temperature until autumn. The culture was exposed to 24 h light, at a light intensity of 20 μmol·m^−2^·s^−1^.

In October, the fertilisation of the spores was induced by decreasing the temperature by 1 °C per day until it reached 10 °C and by exposing the spores to a photoperiod of 12:12. After eight days, the culture was blended for 25 s using a domestic blender and sprayed on collectors for the settlement of the microscopic sporophyte. The collectors were made by cutting a white square drainpipe (section 65 mm^2^) into 50 cm lengths. The hole cutter of an electric drill was used to cut holes on each side of the drainpipe. This facilitates light penetration in the internal part of the strings sprayed with the sporophytes. Sixty meters of polyamide culture string (diameter 1 mm) was wrapped around each collector. A total of 35 m of settlement string allowed for around 30 linear metres on the culture rope at sea. The collectors were then placed in 500 L white fibreglass tanks for 28 days with artificial fluorescent light, at an intensity of 40 μmol·m^−2^·s^−1^, photoperiod 12:12. The seeded string was then coiled around the culture rope at sea, in Bantry Bay (51°39′05.8″ N 9°34′57.6″ W), a large marine inlet, approximately 40 km long. The bay’s greatest depth is approximately 60 m at its mouth. The seabed is predominantly mud to fine sand with increasing medium to coarse sand towards the mouth of the bay. There are areas of medium to coarse sand, coarse sand to gravel, and rock throughout the bay primarily along the perimeter [69]. A total of 11 lines were deployed, parallel to the line coast, between October and December 2019. The lines were numbered progressively with line number, one being the closest to the southern shore and line 11 being the furthest north. The lines were 110 m long and were deployed at a maximum depth of 2 m.

### 3.2. Sampling

Different lines were deployed on different dates and samples were collected on different days during the growing period to assess differences in biomass composition in plants of different ages. Lines 2 and 7 were deployed on 15 October and on 28 November of 2019, respectively and were sampled on 20 March, 3 April, 22 and 30, 5 May and 2 June 2 of 2020. Nine samples were collected from each line, for each sampling date. All samples were hand-cut (30 cm) from the long lines and placed in pre-labelled plastic bags. In the laboratory, samples were shaken 20 times with the aid of a sieve to remove excess water, weighed, and stored at −80 °C. Samples were freeze-dried (Labconco FreeZone 7754030, Kansas City, MI, USA) and milled (<1 mm) with a hand-blender. Samples were then stored at −20 °C for further analysis.

### 3.3. Chemicals

The food-grade enzyme Alcalase 2.4 L was provided by Novozymes (Bagsvaerd, Denmark). 2,2-diphenyl-1-picrylhydrazyl (DPPH), 2,2′-azino-bis(3-ethylbenzothiazoline-6-sulfonic acid) (ABTS), pyrocatechol violet, ferrozine, angiotensin-converting enzyme (ACE) and hippuryl-histidyl-leucine (HHL) were purchased from Sigma-Aldrich (St. Louis, MO, USA). Cu(NO_3_)_2_, Mg(NO_3_)_2_, Mn(NO_3_)_2_, Fe(NO_3_)_3_, NaNO_3_ KNO_3_, Ni(NO_3_)_2_ and Cr(NO_3_)_3_ Pb(NO_3_)_2_ and Cd(NO_3_)_2_ solutions were purchased from Merck (Darmstadt, Germany). All other chemicals were of analytical grade.

### 3.4. Chemical Composition of Alaria Esculenta

The moisture and ash contents were determined according to AOAC methods [70], respectively by drying at 105 °C and combustion of dried samples over 16 h at 500 °C. The nitrogen content was determined using a FP-528 LECO nitrogen analyzer (LECO, St. Joseph, MI, USA) calibrated with EDTA according to the Dumas method [71]. To estimate the total protein content from the analysis of total N, a conversion factor of 5 was used [72]. Total lipid was determined following the method of Bligh and Dyer [73]. Ash determination was performed in triplicate while the other determinations were performed in sextuplicate.

### 3.5. Fatty Acids Profile

The fatty acid methyl esters (FAME) were determined by acid-catalysed transesterification. Briefly, 300 mg of dried sample from the by-products were weighed to a test tube and 5 mL of a 5% acetyl chloride-methanolic solution was added and then left to react for 1 h in a water bath at 80 °C. Once sample extracts were cooled, 1 mL of Milli-Q water and 2 mL n-heptane were added. After a vortex agitation and centrifugation for 3 min at 3000× *g* the organic phase was collected and filtered through anhydrous sodium sulphate. Two millilitres of the final extract were analysed in a GC/FID system consisting of a Bruker Scion 456-GC (West Lothian, UK), equipped with an auto-sampler and with split mode of 100:1. The split/splitless injector and a flame ionisation detector were both set at 250 °C. The separation of the FAME was carried out with helium in a capillary column DB-WAX (Agilent Technologies, Santa Clara, CA, USA) (film thickness, 0.25 μm), 30 m × 0.25 mm i. d., using a temperature program for the column starting at 180 °C and increasing to 200 °C at 4 °C/min, holding for 10 min at 200 °C, heating to 210 °C at the same rate and holding at this temperature for 14.5 min. FAME identification was based on their retention time, using a standard mix (PUFA-3, Menhaden oil, Sigma-Aldrich) as a reference. Results were expressed as a percentage.

### 3.6. Mineral Composition

The elements copper (Cu), magnesium (Mg), manganese (Mn), zinc (Zn), sodium (Na), iron (Fe) potassium (K), nickel (Ni) and chromium (Cr) were measured by flame atomic absorption spectrophotometry (FAAS) (Spectr AA 55B, Varian, Palo Alto, CA, USA) with a background deuterium correction, according to official analytical methods (Jorhem, 2000). Concentrations were determined through linear calibration obtained from absorbance measurements of at least five different concentrations of standard solutions: Cu(NO_3_)_2_, Mg(NO_3_)_2_, Mn(NO_3_)_2_, Fe(NO_3_)_3_, NaNO_3_ KNO_3_, Ni(NO_3_)_2_ and Cr(NO_3_)_3_ (1 g/L dissolved in 0.5 M HNO_3_). Detection limits (DL) were calculated from the residual standard deviation of the response and the slope of the calibration curve of each element. The values obtained were 0.01 (K), 0.09 (Na), 0.02 (Mg), 0.06 (Zn), 0.32 (Fe), 0.02 (Cu), 0.01 (Mn), 0.02 (Ni) and 0.09 (Cr) mg kg^−1^ wet weight. All data are reported in mg kg^−1^ on a wet weight basis.

### 3.7. Contaminants

Cadmium (Cd) and lead (Pb) were determined by graphite furnace atomic absorption spectrometry (GF-AAS), using an Agilent apparatus Spectr 240Z (Santa Clara, CA, USA) with a Zeeman correction. The methodology followed was based on the European Standard EN 14084 (CEN, 2003). The concentrations were determined through linear calibration obtained from absorbance measurements of, at least, five different concentrations of standard solutions: Pb(NO_3_)_2_ and Cd(NO_3_)_2_ (1 g/L in 0.5 M HNO_3_).

The quantification of total mercury (HgT) was done by atomic absorption spectrophotometry (AAS-DMA) in a direct mercury analyser spectrophotometer (AMA 254, Leco, St. Joseph, MI, USA) according to EPA (2007). The analyses were performed in triplicate. All data are reported in mg kg^−1^ on a wet weight basis.

Detection limits were 0.002 (Cd), 0.06 (Pb) and 0.004 (Hg) mg kg^−1^ wet weight. Certified reference material (Dolt5, Dog fish liver, National Research Council of Canada, Ottawa, ON, Canada) was used to control the accuracy of the results.

Detection limits were 0.002 (Cd), 0.06 (Pb) and 0.004 (Hg) mg kg^−1^ wet weight. Certified reference material (Dolt5, Dog fish liver certified reference material for trace metals and other constituents: National Research Council of Canada, Ottawa, ON, Canada) to assess analytical method accuracy for all determined elements results obtained in this work were in good agreement with the certified values.

### 3.8. Seaweed Extracts Preparation

Enzyme-assisted extractions with Alcalase (EAA) were performed according to the method described by Sapatinha et al. [58]. Briefly, dried seaweed was mixed with distilled water and the pH of the mixture was adjusted to 8.0. Alcalase (150 mg) was added, and the mixture was incubated at 50 °C for 24 h. Then, the enzyme was inactivated by heating at 100 °C for 10 min and the mixture was centrifuged, filtered and freeze-dried.

### 3.9. Extraction Yield

The extraction yield of the different methods used was calculated as the weight percentage of lyophilised extract to the weight of dried seaweed submitted to extraction.
$$Extraction\;yield\;\left(\%\right)=\frac{dry\;extract\;\left(g\right)}{dry\;seaweed\;\left(g\right)}\;\times 100$$

### 3.10. Total Phenolic Content (TPC)

Total phenolic content (TPC) was determined using the colourimetric method of Folin–Ciocalteau using gallic acid as standard (5–500 mg/mL). Briefly, 150 μL of seaweed extract solutions (3–5 mg/mL) were mixed with 750 µL of 1:10 diluted Folin–Ciocalteu reagent and after 4 min at room temperature, 600 μL of 10% sodium carbonate was added. The absorbance of the mixture at 765 nm was measured after 2 h of incubation at room temperature in the dark. The blank was prepared using distilled water instead of the seaweed extract solution. The results were expressed as mg of gallic acid equivalent (GAE)/g dry weight of seaweed extract. All analyses were made at least in triplicate and the results are presented as mean value ± SD.

### 3.11. Total Flavonoids Content (TFC)

Total flavonoid content (TFC) was determined according to the spectrophotometric method described by Pekal et al. [74]. In short, 0.5 mL of 2% AlCl_3_ was added to 1 mL of the sample (1–5 mg/mL) and then 0.5 mL of water was added. The mixture was vigorously shaken and then after 10 min of incubation at room temperature, 4 mL of water was added and then the absorbance was measured at 425 nm. In the blank, the amount of AlCl_3_ solution was substituted by the same amount of water. Quercetin (25–250 μm) was used as standard and the results were expressed as mg of quercetin equivalent (QE)/g dry weight of seaweed extract. All analyses were made at least in triplicate and the results are presented as mean value ± SD.

### 3.12. Antioxidant Activity

#### 3.12.1. 2,2-Diphenyl-1-picrylhydrazyl (DPPH) Radical Scavenging Activity

The determination of DPPH radical scavenging activity was done according to the method of Rodrigues et al. [56]. Briefly, 100 µL of the different seaweed extracts (1–20 mg/mL) were added and mixed with 1.0 mL of 0.1 mM DPPH solution in 95% ethanol in an *Eppendorf* tube.

The mixture was vortexed for 1 min and then placed in a water bath (30 °C) for 30 min in the dark, thereafter samples were centrifuged at 10,000× *g* for 5 min. The absorbance of the solution was measured at 517 nm using an Evolution 201 UV-Visible Spectrophotometer (Thermo Scientific, Waltham, MA, USA). The control was prepared in the same way except that distilled water was used instead of the sample solution. The radical scavenging activity of DPPH was calculated by the percentage inhibition of DPPH as follows:$$DPPH^{•}\;\mathrm{scavenging}\;\mathrm{activity}\;\left(\%\right)=\frac{A_{C}-A_{S}\;}{A_{C}}\times 100$$
where *A_S_* and *A_C_* correspond to the absorbance of the sample and control, respectively. All analyses were made at least in triplicate and the results are presented as mean values. The EC_50_ value was calculated for each seaweed extract.

#### 3.12.2. 2,2′-Azino-bis(3-ethylbenzothiazoline-6-sulfonic acid) (ABTS) Radical Scavenging Activity

The ABTS radical scavenging activity of FPH was performed according to Re et al. (1999). ABTS radical cation ABTS^•+^ was prepared with a final concentration of 7 mM ABTS in 2.45 mM potassium persulfate. This mixture was kept in the dark at room temperature for 16 h before use. ABTS^•+^ solution was diluted with 5 mM sodium phosphate buffer (pH 7.4) to obtain an absorbance value of 0.70 ± 0.02 at 734 nm. A 20 µL of solution at different seaweed extract concentrations (0.5–20 mg/mL) was mixed with 2 mL of ABTS^•+^ solution and then incubated in the dark at 30 °C (water bath) for 6 min. The absorbance values of the mixture were read at 734 nm using an Evolution 201 UV-Visible Spectrophotometer (Thermo Scientific, Waltham, MA, USA). The control was prepared in the same manner using distilled water instead of the sample solution. All determinations were made at least in triplicate and the EC_50_ value was calculated for each seaweed. The ABTS radical scavenging activity was calculated according to the following equation:$$ABTS^{•+}\;\mathrm{scavenging}\;\mathrm{activity}\;\left(\%\right)=\frac{A_{C}-A_{S}\;}{A_{C}}\times 100$$
where *A_C_* represents the absorbance of the control and *A_S_* represents the absorbance of the sample.

#### 3.12.3. Reducing Power

The reducing power was determined following Oyaizu’s method described by Parthiban et al. [75]. Briefly, 1 mL of seaweed extract solutions with different concentrations (1–20 mg/mL) was mixed with 2.5 mL phosphate buffer (2 M, pH 6.6) and 2.5 mL potassium ferricyanide (1%). The reaction mixture was kept in a water bath at 50 °C for 20 min, 2.5 mL of TCA (10%) was added, and the mixture was centrifuged at 1500× *g* for 10 min. Finally, 2.5 mL of the supernatant was mixed with 2.5 mL distilled water and 0.5 mL of 0.1% ferric chloride and the absorbance was measured after 15 min incubation in the dark at 700 nm (Evolution 201 UV-Visible Spectrophotometer, Thermo Scientific). The control was prepared using all reagents with distilled water instead of the sample solution. All analyses were carried out at least in triplicate. The concentration for the absorbance value of 0.5 value was determined for each extract.

### 3.13. Metal Chelating Activities

#### 3.13.1. Cu^2+^ Chelating Activity

The copper chelating activity was evaluated by copper chelate titration using pyrocatechol violet (PV) as the metal chelating indicator with slight modifications as described by Torres-Fuentes et al. [76]. In short, 1 mL of 0.1 mg/mL CuSO_4_ in 50 mM sodium acetate buffer pH 6.0 was mixed with 1 mL of sample solution prepared at different concentrations (0.5–20 mg/mL). Then, 250 µL of PV 0.3 mM in 50 mM sodium acetate buffer pH 6.0 was added and the PV + Cu^2+^ complex was formed. The absorbance was read at 626 nm (Evolution 201 UV-Visible Spectrophotometer, Thermo Scientific). The control was prepared in the same way by using distilled water instead of the sample solution. All determinations were carried out at least in quadruplicate and the EC_50_ was estimated for each seaweed extract. The chelating activity was calculated using the following formula:$${\mathrm{Cu}}^{2+}\;\mathrm{chelating}\;\mathrm{activity}\;\left(\%\right)=\frac{A_{C}-A_{S}\;}{A_{C}}\times 100$$
where *A_S_* and *A_C_* correspond to the absorbance of the sample and control, respectively.

#### 3.13.2. Fe^2+^ Chelating Activity

The iron chelating activity of the seaweed extracts was estimated by the method described by Decker and Welch [77]. Briefly, to 1 mL of each sample solution prepared at different concentrations (1–20 mg/mL), 3.7 mL of distilled water and 100 µL of 2 mM ferrous chloride were added and mixed. Then, the reaction was initiated by the addition of 200 µL of 5 mM ferrozine solution and the mixture was vortexed and kept at room temperature for 10 min. The absorbance of the resulting solution was read at 562 nm (Evolution 201 UV-Visible Spectrophotometer, Thermo Scientific). The control was prepared in the same way by using distilled water instead of the sample solution. All determinations were carried out at least in quadruplicate and the EC_50_ value was determined. The percentage of inhibition of ferrozine Fe^2+^ complex formation was calculated by the formula:$${\mathrm{Fe}}^{2+}\;\mathrm{chelating}\;\mathrm{activity}\;\left(\%\right)=\frac{A_{C}-A_{S}\;}{A_{C}}\times 100$$
where *A_C_* is the absorbance of ferrozine + Fe^2+^ complex in the absence of the extract sample and *A_S_* is the absorbance of ferrozine + Fe^2+^ complex in the presence of the extract sample.

### 3.14. Anti-Hypertensive Activity—ACE Inhibitory Activity

The ACE inhibitory activity using Hippuryl-L-Histidyl-L-Leucine (HHL) as substrate was evaluated by high performance liquid chromatography (HPLC). Briefly, 10 µL of seaweed extract solution (5 mg/mL), 10 µL of borate buffer (used as blank), or captopril 0.0217 mg/mL (standard inhibitor) were mixed with 10 µL of 0.2 U/mL ACE. The mixture was pre-incubated at 37 °C for 20 min and after this 50 µL of HHL was added and the mixture was incubated at 37 °C for 30 min. The reaction was stopped by the addition of 85 µL of 1 M HCl, the solution was filtered and an aliquot (10 μL) was injected into a HPLC HP Agilent 1050 series (Agilent, Santa Clara, CA, USA) equipped with a reversed-phase C18 column (100 mm × 4.6 mm, 2.6 m, 100 Å; Kinetex Phenomenex). The identity of hippuric acid (HA) and HHL was assessed by comparison with the retention times of standards. The peak areas were obtained with the software Agilent ChemStation for LC (Agilent, Santa Clara, CA, USA) and the percentage of ACE inhibition was calculated as follows:$$ACE\;inhibition\;\left(\%\right)=\frac{HA_{buffer}-HA_{sample}\;}{HA_{buffer}}\times 100$$
where *HA_buffer_* is the concentration of HA in the reaction with the buffer instead of the sample, and *HA_sample_* is the concentration of HA in the reaction with the sample. Captopril showed an ACE inhibitory activity of ca. 95–100%.

### 3.15. Statistical Analysis

All statistical analyses were performed using the software STATISTICA© version 12 (data analysis software system) from StatSoft, Inc. (Tulsa, OK, USA). The results of the analyses are reported as mean values ± standard deviation (SD) and differences between mean values were performed using a one-way analysis of variance (ANOVA). For this, the Tukey test was applied with a significance value of *p* < 0.05.

A multivariate analysis on antioxidant activities (DPPH, ABTS, reducing power), chelating activities (Fe^2+^ and Cu^2+^), and anti-hypertensive activities were performed by principal components analysis and the two main factors were represented.

## 4. Conclusions

The study allowed the identification of an optimal ‘growing window’ to produce high-quality biomass. There was a specific focus on the identification of the best months for deployment (in autumn) and the best months for harvesting (in spring/summer). The growth analysis highlighted that an early deployment allows to maximise production. Lines deployed in October gain an advantage in biomass production that lines deployed a month later are not able to recuperate. An increase in biomass in spring is not as effective as in autumn and in winter. This decreased production is not due to the increased temperature in spring, as the temperature in October and November was higher than in March. Better growth in early deployed lines could be linked to endogenous rhythms [78]. Another possibility is that the reduction in pigments in spring causes a reduced ability to gain biomass.

Regarding the harvesting time, this study highlights that early harvesting favours qualitative parameters. The biomass sampled earlier in the year had higher protein and lipid content and higher phenolics and flavonoids that conferred a higher antioxidant activity to it. A late harvest was also linked to erosion and epiphytic contamination.

The study confirms the health benefits associated with the inclusion of seaweed in the diet. *A. esculenta* contains low SFA and high PUFA and MUFA, recommended for the health of the cardiovascular system. A high content of EPA and AA and a high omega3/omega6 ratio were also found. *A. esculenta* is also a good source of minerals, while accumulation of heavy metals was not observed in the samples analysed in this study.

Based on our research, *A. esculenta* is a distinguished source of various bioactive compounds with abundance in many minerals and could be utilised as novel functional foods which provide health-beneficial activities. Therefore, it is not surprising that this seaweed showed the highest potential for use as an ingredient in functional foods and nutraceuticals due to its evaluated biological activities and high-value compounds.

## Figures and Tables

**Figure 1 marinedrugs-21-00305-f001:**
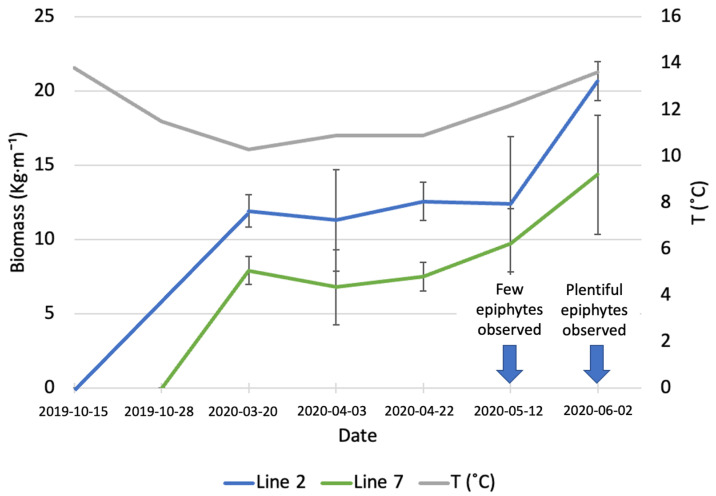
Increase in biomass of *A. esculenta* produced in line 2 (deployed on 15 October 2019) and line 7 (deployed on 28 November 2019) through the growing season.

**Figure 2 marinedrugs-21-00305-f002:**
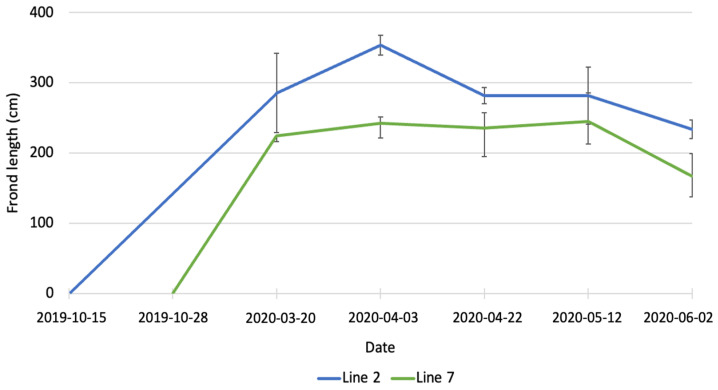
Increase in length of *A. esculenta* produced in line 2 (deployed on 15 October 2019) and line 7 (deployed on 28 November 2019) through the growing season.

**Figure 3 marinedrugs-21-00305-f003:**
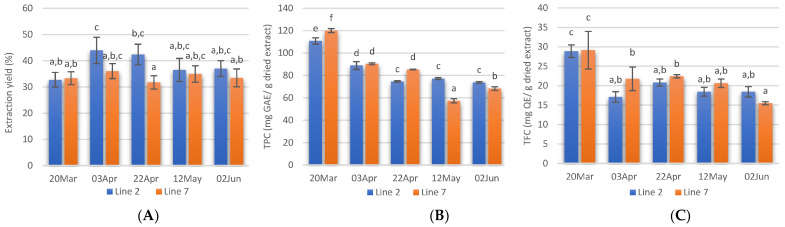
Extraction yields (**A**), total phenolic content (**B**), and total flavonoids content (**C**) of extracts prepared with *A. esculenta* deployed on 15 October(line 2) and on 28 November (line 7) and harvested on 20 March, 3 April, 22 April, 12 May and 2 June of 2020. Means denoted by a different letter (a, b,…) indicate significant differences (*p* < 0.05).

**Figure 4 marinedrugs-21-00305-f004:**
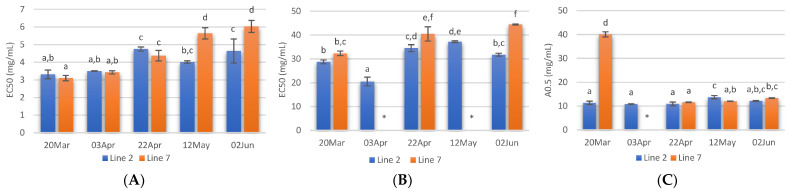
ABTS (**A**) and DPPH (**B**) radical scavenging activity activities and reducing power (**C**) of extracts prepared with *A. esculenta* deployed on lines 2 and 7 and sampled on 20 March, 3 April, 22 April, 12 May and 2 June of 2020. * did not attain 50% inhibition. Means denoted by a different letter (a, b,…) indicate significant differences (*p* < 0.05).

**Figure 5 marinedrugs-21-00305-f005:**
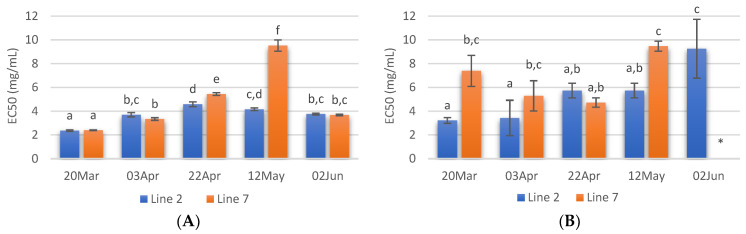
Cu^2+^ chelating activity (**A**) and Fe^2+^ chelating activity (**B**) of extracts prepared with *A. esculenta* deployed on lines 2 and 7 and sampled on 20 March, 3 April, 22 April, 12 May and 2 June 2020. * did not attain 50% inhibition. Means denoted by a different letter (a, b,…) indicate significant differences (*p* < 0.05).

**Figure 6 marinedrugs-21-00305-f006:**
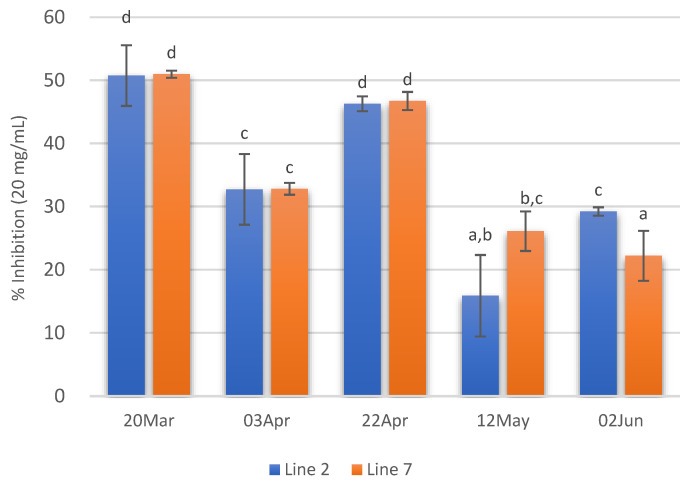
ACE inhibition of extracts (20 mg/mL) prepared with *A. esculenta* deployed on lines 2 and 7 and sampled on 20 March, 3 April, 22 April, 12 May and 2 June 2020. Means denoted by a different letter (a, b,…) indicate significant differences (*p* < 0.05).

**Figure 7 marinedrugs-21-00305-f007:**
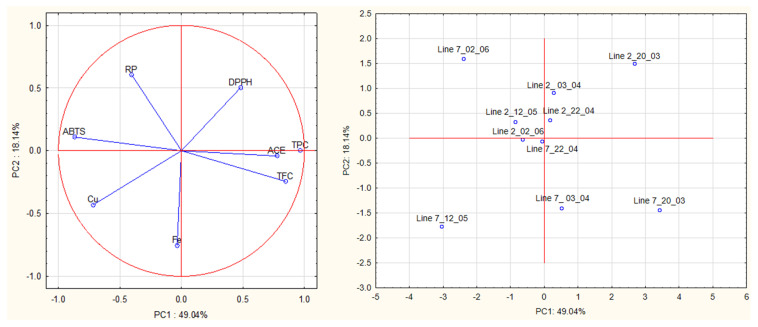
Principal component analysis (PCA) of antioxidant activities (DPPH, ABTS, reducing power), chelating activities (Fe^2+^ and Cu^2+^), and anti-hypertensive activities of extracts prepared with *A. esculenta*.

**Table 1 marinedrugs-21-00305-t001:** Proximate composition (g/100 g) of dried *Alaria esculenta* deployed on 15 October and 28 November.

	Line 2 (Deployed 15 October 2019)	Line 7 (Deployed 28 November 2019)
Moisture	8.6 ± 0.02 ^a^	9.5 ± 0.01 ^b^
Protein	11.8 ± 0.2 ^b^	11.2 ± 0.2 ^a^
Lipids	2.3 ± 0.2 ^b^	1.8 ± 0.2 ^a^
Ash	32.28 ± 0.03 ^b^	32.00 ± 0.02 ^a^
Other components ^#^	42.1	42.7

^#^ Calculated by difference. Means denoted by a different letter (^a^, ^b^,…) indicate significant differences (*p* < 0.05) between *A. esculenta* samples from the lines deployed on the 15 October and the 28 November.

**Table 2 marinedrugs-21-00305-t002:** Saturated (SFA), monounsaturated (MUFA) and polyunsaturated (PUFA) fatty acids, expressed as relative % of total fatty acid and ω3/ω6 ratio. Major fatty *acids* in dried *Alaria esculenta* are also reported in the table and expressed as % of total fatty acids.

		Line 2 (Deployed 15 October 2019)	Line 7 (Deployed 28 November 2019)
Fatty acids types	SFA	30.10 ± 3.75	29.71 ± 2.44
	MUFA	14.37 ± 2.45	13.89 ± 1.59
	PUFA	51.36 ± 6.05	52.37 ± 3.4
	ω3/ω6	2.11 ± 0.24	2.19 ± 0.23
Major fatty acids	Palmitic acid (16:0)	15.68 ± 3.39	15.19 ± 2.54
	Oleic acid (18:1 ω9)	11.17 ± 1.67	11.31 ± 0.82
	Arachidonic acid (20:4 ω6)	11.07 ± 1.71	10.80 ± 1.27
	EPA (20:5 ω3)	10.03 ± 1.23	10.10 ± 0.56
	DHA (22:6 ω3)	0.65 ± 0.99	0.81 ± 1.37

No significant differences between *A. esculenta* samples deployed on 15 October and 28 November.

**Table 3 marinedrugs-21-00305-t003:** Mineral composition (mg/kg dw) of dried *A. esculenta* and DRI for adults.

	Line 2 (Deployed 15 October 2019)	Line 7 (Deployed 28 November 2019)	DRI (mg/day) ^#^
K	80,106 ± 5090 ^b^	66,595 ± 2506 ^a^	4700
Na	54,468 ± 4052 ^a^	59,077 ± 12,077 ^a^	1500
Mg	9033 ± 100 ^a^	8992 ± 567 ^a^	310
Fe	397.3 ± 28.5 ^b^	23.5 ± 0.0 ^a^	8.0
Zn	28.6 ± 0.5 ^b^	1.29 ± 0.06 ^a^	8
Cu	1.56 ± 0.01 ^b^	9.29 ± 0.04 ^a^	0.9
Mn	11.10 ± 0.35 ^b^	0.78 ± 0.04 ^a^	1.8
Cr	1.02 ± 0.04 ^b^	0.81 ± 1.37 ^a^	0.025
Ni	0.67 ± 0.02 ^b^	0.51 ± 0.01 ^a^	0.5

^#^ DRI—dietary reference intakes estimated for adults [49]. Values are presented as average ± standard deviation. Means denoted by a different letter (^a^, ^b^,…) indicate significant differences (*p* < 0.05) between *A. esculenta* samples from the lines deployed on 15 October and on 28 November.

**Table 4 marinedrugs-21-00305-t004:** Cadmium, lead and mercury levels (mg/kg) of dried *A. esculenta*.

	Line 2 (Deployed 15 October 2019)	Line 7 (Deployed 15 October 2019)
Cd	0.72 ± 0.15	0.75 ± 0.16
Pb	1.1 ± 0.1	1.1 ± 0.1
Hg	<LQ	<LQ

No significant differences between metals content in samples of *A. esculenta* from line 2 and 7. LQ—limit of quantification, 0.011 mg/kg.

## Data Availability

The datasets generated for this study are available on request to the corresponding author.

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
