# Peer review of "Effect of Deployment and Harvest Date on Growth and High-Value Compounds of Farmed Alaria esculenta"

_marinedrugs, 2023, doi:10.3390/md21050305_

Round 1

Reviewer 1 Report

This paper reported the effect of deployment and harvest date on growth and high-value compounds of farmed Alaria esculenta. The bioactivity was also evacuated and compared. Overall, the manuscript is well-organized and written, and the manuscript should be published after major revisions.

1.     For the help of reading, please give the full description of the abbreviations, e.g., PUFA.

2.     In part 2.2, please clarify in the text when lines 2 and 7 were deployed. Why lines 2 and 7, not other lines?

3.     “however, this increase was statistically significant only for line 2 (p=0.3).” this is not statistically significant.

4.     Figure 1, the quality should be improved. The labels of the y-axis are too small; Fix the x-axis lable “15Oct1”; For the help of reading, the x-axis could be labeled as 2019-10-15, for example. Same as Figure 2.

5.     Table 1, the letters “a” and “b” are a little bit confusing. Please make it clear. What’s the significant difference?

6.     Table 2, there is just one letter “a”; For the % of each component, is there just “+”? Please double-check the number; If you add all the component % together, it will be more than 100%, for example, line 2; how do you explain that?

7.     Table 3, what the meaning of “b”?

8.     Figure 3 is hard to read. There are panels a, b, and c; but also have letters a, b, c, and d above the column. What’s the meaning of each letter? For panel (c), what happened to the average value bar, like 20Mar, 03Apr and 02Jun? Similar problems with Figures 4 and 5.

9.     4.7. Extraction yield”  should be part “4.8”.

Reviewer 2 Report

The manuscript of Blanco et al., investigated the seasonal variability on the growth and composition of a brown seaweed species (Alaria esculenta) in a European context. The use of seaweed is a widely studied topic that is highly relevant, especially in today’s global situation (potential climate mitigating feedstock) and the fact that seaweed cultivation in Europe is gaining momentum. Therefore this paper deserves recognition in this field of research. The approach of the authors is well performed, the article is well written and the applied methodology is excellent. Therefore this manuscript deserves publication in Marine Drugs. However, after reading the manuscript I have some minor comments which are ought to be addressed before publication:

-       -  Line 24: “The content of Cd, Pb Hg was relatively low and above the maximum levels allowed.” Should tis not be “below”, based on the results presented?

-     -    Please mind significant numbers in the text and be consistent throughout the text (for instance in the abstract section and Table 4)

-     -    Authors use both seaweed and macroalgae in terms of terminology. This is correct of course, but for consistency reasons please pick one and use that one throughout the text.

-     -    In section 3.2, please already indicate what is meant with “extraction yield”. It is mentioned in the materials and methods section (later on in the manuscript), but to facilitate reading the reviewer recommends to already give some information when first mentioned in the text. For instance by placing “solubilized seaweed” between brackets.

-     -    Please add an explanation of the indices a,b,c,d,e and f in the caption of Figure 4 and 5.

-     -    Typo: superscript 2 in mm² in Line 426

-     -    Is there a specific reason why the authors didn’t measure Calcium?

-    -     The samples were hand-blend prior analysis. Have the authors any idea about the particle size or particle size distribution? Please add.

-     -    Please add a “Chemicals” section in the materials and methods section that lists all chemicals used and where they are bought from (especially for Alcalase this is highly significant).

-      -   Please also mind the spacing between number and unit throughout the text (consistency).

Round 2

Reviewer 1 Report

The authors fixed some of the problems, however, there are still some that need to be revised.

Major comments:

1.     In Tables 1-3, the authors said, “Different letters in each line indicate significant differences (p<0.05)”, do “a” and “b” represent the exact p-value? If so, why use two different letters? Please make it clear.

2.     In Figures 3-6, what do the different letters (a,b,c,d,e,f) mean on each column? Please clarify in detail either in the footnote or main text.

Round 3

Reviewer 1 Report

The manuscript is suitable for publishing now.